# Legible Robot Motion from Conditional Generative Models

**Matthew Bronars** [1]   **Danfei Xu** [1]

## Abstract

In human robot collaboration, legible motion that clearly conveys its intentions and goals is essential. This is because forecasting a robot's next move can lead to an improved user experience, safety, and task efficiency. Current methods for generating legible motion utilize hand designed cost functions and classical motion planners, but there is need for data driven policies that are trained end-to-end on demonstration data. In this paper we propose Generative Legible Motion Models (GLMM), a framework that utilizes conditional generative models to learn legible trajectories from human demonstrations. We find that GLMM produces motion that is $76\%$ more legible than standard goal conditioned generative models and $83\%$ percent more legible than generative models without goal conditioning.

## 1. Introduction

As robots become more integrated into our daily lives, it is critical that they move in a way that is not only efficient and functional, but also legible and understandable to humans. In Human-Robot Interaction (HRI), legible motion conveys the robot's intentions and goals in an intuitive and interpretable manner (Dragan et al., 2013; Lichtenthäler et al., 2011). Making a robot's actions more transparent will allow humans to better anticipate and respond to the robot. This can reduce the risk of accidents and collisions, which is important in safety critical environments. Studies have also shown that in collaborative environments, being able to forecast a robot's motion leads to faster task completion times and more fluent collaboration (Breazeal et al., 2005; Dragan et al., 2015). Mathematically, a legible trajectory is one that maximize $P(g^*|\xi_{s\rightarrow q})$ where $g^*$ is the goal and $\xi_{s\rightarrow q}$ is the ongoing trajectory.

---

*Equal contribution [1]Department of Computer Science, Georgia Institute of Technology, Atlanta, Georgia, USA. Correspondence to: Matthew Bronars <mbronars@gatech.edu>, Danfei Xu <danfei@gatech.edu>.

*Proceedings of the 40th International Conference on Machine Learning*, Honolulu, Hawaii, USA. PMLR 202, 2023. Copyright 2023 by the author(s).

Traditional methods for generating legible motion leverage hand designed cost functions and classical motion planning algorithms (Dragan et al., 2013). For simple tasks, these cost functions and motion planners are well explored and relatively easy to implement. Experiments have shown that human observers expect robots to follow minimum distance paths when reaching for objects (Dragan & Srinivasa, 2014). Motion planners are able to maximizes $P(g^*|\xi_{s\rightarrow q})$ by utilizing this cost function. While effective for pick and place, as task complexity increases, it becomes harder to explicitly model cost functions and the desired dynamics. For this reason, a data driven approach that directly learns legible motion from demonstration data is desirable.

In this work, we aim to show that conditional generative models are ideal for producing legible motion for complex manipulation tasks. Generative models can capture distributions for highly unstructured data such as natural images and conditionally generate samples with class-specific characteristics (Dhariwal & Nichol, 2021). We argue that in the action generation domain, these class specific characteristics are the same characteristics that make a trajectory legible. This is an important insight because probabilistic generative models such as GANS (Goodfellow et al., 2020), VAEs (Kingma & Welling, 2013), and Diffusion Models (Rombach et al., 2022) lend themselves particularly well to HRI. Generative models effectively capture data multi-modality, which is important because humans are multi-modal actors and demonstrate many different ways to accomplish a task. Legible motion is about more than just reaching a goal, it is tied to the *way* in which you reach the goal. Thus, learning legible trajectories from demonstrations necessitates modeling as many possible modes in the data, and then selecting actions from the one that is most legible.

Our method, GLMM (Generative Legible Motion Models), is an end-to-end framework for generating legible motion from multi-modal human demonstration data. The key idea is to formulate legible motion planning as a conditional generative modeling problem, where the conditions are specified by a motion legibility classifier. We assume a multi-task demonstration data, where the demonstrated behaviors cover a variety of ways to reach any given goal, akin to the setting studied in (Grauman et al., 2022; Lynch et al., 2020). One of these modes will be more legible than the rest, and we

learn a generative policy that produces actions from this mode. By jointly training a classifier and generative model, we are able to rank sub-goals produced by the model. This implicit classifier guidance allows us to follow a trajectory that optimizes for legibility.

For our preliminary experiments, we train GLMM on 280 demonstrations of a block reaching task with two offset goals. This allows us to evaluate the generated trajectories using functions that are known to measure legibility for pick and place tasks. Two main conclusions can be drawn from our work: Conditioning through classifier guided produces actions that are 76% more legible than standard goal - conditioned models and 83% percent more legible than unconditioned models. Secondly, there is a trade-off between legibility and task success rate. We hope our results ensure that legibility continues being incorporated into robot policies as the field moves towards deep learning approaches.

## 2. Previous Work

### 2.1. Legible Robot Motion

Shared intentionally is an important aspect of human cognition, and being able to read intentions is critical for how we collaborate as a species (Tomasello et al., 2005). Intent expressive actions (legible actions) are a form of non-verbal communication that allows groups of agents to coordinate their behaviors. This is useful for HRI because if robots forecast their next move, they can fluidly interact and improvise with humans (Hoffman & Weinberg, 2010). Robots are more readable and understandable if they have the capability to express forethought and respond to task outcomes. This increases people's perception of robots and will make users more willing to engage in interactions with legible robots (Takayama et al., 2011). Similarly, motion produced by robots can be legible if it allows for quick and confident predictions of the goal state. Experiments have shown that legible motion allows for faster completion time of collaborative tasks and increased user satisfaction (Dragan et al., 2015). In this paper we focus specifically on generating legible motion trajectories.

Standard methods for generating legible motion involve hand designed cost functions as described in section 3.1. With these cost functions, classical motion planners such as Covariant Hamiltonian Optimization (Zucker et al., 2013) are able to generate legible trajectories. These trajectories maximize the relative cost to reach goals that are not the target goal. While effective for simple tasks with well defined cost functions, these methods may fall short as robots take over increasingly complex tasks in our society. For this reason, finding ways for deep learning based policy generators to produce legible motion is an important task.

Recently, generating legible motion has been accomplished using reinforcement learning algorithms. In (Busch et al., 2017), direct policy search makes iterative improvement to the parameters of a dynamical movement primitive (DMP). The cost function for this method incorporates motion smoothness, time to prediction, and accuracy of prediction. While this algorithm can be applied to complex tasks, it relies on time consuming data collection and classical methods for motion generation. The authors of (Zhao et al., 2020) introduce an actor critic approach for legible motion generation. In their algorithm the actor is a motion planner and the critic predicts what the next 20 action steps will be. These networks are jointly trained, and the reward for the actor is tied to how close its actions are to the critic's predictions. While this method does not require hand designed cost functions or classical motion planners, GLMM improves in two main ways. First, our method learns from offline data, which is safer when deploying around humans as it does not require online interaction of a partially trained agent. Second, the actor critic approach assumes that the critic's predicted actions will align with an observer's predicted actions. GLMM relaxes this assumption because it directly imitates the data. We only assume that a neural network can identify which of the demonstrated trajectories is most legible.

### 2.2. Learning from Multi-Modal Demonstrations

Learning effective policies from demonstration data is an important open problem in robotics. There are many benefits to learning from large scale, offline data such as scalability, portability, and reproducability. These factors are particularly important for deep learning and are part of the reason we see state of the art performance from deep vision and language models (Deng et al., 2009; Krizhevsky et al., 2017; Devlin et al., 2018; Floridi & Chiriatti, 2020). While we are not able to train policy generators that are similarly effective, learning from demonstration data is still a commonly used technique. The two main paradigms for learning from offline demonstrations are Imitation Learning (Pomerleau, 1988; Zhang et al., 2018; Mandlekar et al., 2020b) and Batch (Offline) Reinforcement Learning (Levine et al., 2020; Lange et al., 2012; Cabi et al., 2019). These algorithms assume access to datasets of state action pairs and reward labels in the case of offline RL. Most formulations of imitation learning assume access to expert demonstrations, but empirical studies have gotten state of the art performance across a variety of tasks even with sub-optimal data (Mandlekar et al., 2021; Florence et al., 2022; Hahn et al., 2021).

In the context of HRI, learning from demonstrations provides a whole other set of benefits. LfD allows for non-expert programming of desired behaviors through kinesthetic teaching, teleoperation, or passive observation (Ravichandar et al., 2020). Because of this, fine-tuning

by end users is much simpler and allows for greater adaptability. This is particularly useful when training generative models as they have the capacity for continual learning through techniques like deep generative replay (Shin et al., 2017). Another important factor is the safety offered by offline learning. Deep reinforcement learning algorithms have achieved excellent results across a wide range of domains, but they necessitate online interaction with the environment (Levine et al., 2020). Deploying a partially trained agent can be dangerous because actions with low reward (such as hitting a human) may still be taken. When using LfD algorithms this danger can be mitigated by only deploying fully trained agents, even if they are going to be fine-tuned by end users.

For this paper, we are specifically concerned with learning multi-modal action distributions from demonstrations of robot manipulation tasks. These distributions don't have a singular deterministic action output, rather there can be multiple plausible actions from any given state. While hard to optimize for explicitly, a number of imitation learning algorithm have been successful at replicating multi-modal training data. Recently, Behavior Transformers have successfully imitated robot manipulation tasks while capturing the major modes present in the data (Shafiullah et al., 2022). Various papers have successfully used diffusion models, a class of generative model, to learn multi-modal policies and flexibly synthesize behavior (Chi et al., 2023; Janner et al., 2022; Ajay et al., 2022). VAEs are another class of generative models that are particularly effective in generating multi-modal data (Mandlekar et al., 2020a; 2021). Our method, GLMM, is an extension of VAEs that incorporates implicit classifier guidance.

# 3. Preliminaries

## 3.1. Equations for Legible Motion

Mathematically, a legible trajectory $\xi$ from start state $s$ to goal state $g^*$ optimizes the following equation (Dragan et al., 2013):

$$legibility(\xi) = \frac{\int P(g^*|\xi_{s \to \xi(t)})f(t)dt}{\int f(t)dt} \quad (1)$$

Here $f(t)$ is a function of time that puts higher weight on earlier parts of the trajectory. Typically, $P(g|\xi_{s \to q})$ is computed using a cost function $C$ that models what the observer expects the robot to do:

$$P(g|\xi_{s \to q}) \propto \frac{exp(-C[\xi_{s \to q}] - V_g(q))}{exp(-V_g(s))} P(g) \quad (2)$$

Here $V_Y(X)$ is the lowest cost path from $X$ to $Y$. In order to maximize $P(g^*|\xi_{s \to q})$, one must minimize $P(g \neq$

$g^*|\xi_{s \to q})$. This is done by following an ongoing path $\xi_{s \to q}$ such that $V_{g \neq g^*}(q) \gg V_{g^*}(q)$. For pick and place tasks, experiments have shown (Dragan & Srinivasa, 2014) that the cost function C is:

$$C[\xi] = \frac{1}{2} \int \xi'(t)^2 dt \quad (3)$$

From this equation, it is clear that longer, slower paths have higher cost. So straight line paths that move quickly towards an object have minimum cost for pick and place tasks. In this paper, we draw inspiration from these equations when evaluating the legibility of the trajectories our model produces.

## 3.2. Sequential Decision Making

We view robot action generation as a sequential decision making problem and model it as a discrete-time infinite-horizon Markov Decision Process (MDP), $\mathcal{M} = (S, A, T, R, \gamma, \rho_0)$, where $S$ is the state space, $A$ is the action space, $T(\cdot|s, a)$ is the state transition distribution, $R(s, a, s_0)$ is the reward function, $\gamma \in [0, 1)$ is the discount factor, and $\rho_0(\cdot)$ is the initial state distribution. At every step, an agent observes a state $s_t$ and queries a policy $\pi$ to choose an action $a_t = \pi(s_t)$. The agent performs the action and observes the next state $s_{t+1} \sim T(\cdot|s_t, a_t)$ and reward $r_t = R(s_t, a_t, s_{t+1})$.

We augment this MDP with a set of absorbing goal states $G \subset S$, where $g \in G$ corresponds to a specific state of the world in which the task is considered to be solved. Every pair $(s_0, G)$ of an initial state $s_0 \sim \rho_0(\cdot)$ and goals for a task $G$ corresponds to a new task instance.

We assume access to a dataset of $N$ task demonstrations $D = \{\tau_i\}_{i=1}^N$ where each demonstration is a trajectory $\tau_i = (s_{i0}, a_{i0}, s_{i1}, a_{i1}, \ldots, s_{iT})$ that begins in a start state $s_{i0} \sim \rho_0(\cdot)$ and terminates in a final (goal) state $s_{iT} = g_i$.

## 3.3. Classifier Guidance

Generative algorithms model $P(s|g)$ where $s$ is the state and $g$ is the goal. From Bayes' Rule we can show that with access to a classifier $P(g|s)$, we are able to turn an unconditional model $P(s)$ into a conditional model $P(s|g)$.

$$P(s|g) = \frac{P(g|s) * P(s)}{P(g)} \quad (4)$$

$$\implies \log P(s|g) = \log P(g|s) + \log P(s) - \log P(g) \quad (5)$$

This is frequently leveraged in diffusion models in the form of classifier guidance. Conditional diffusion models estimate $\Delta_s \log P(s|g)$. When implementing classifier guidance (Dhariwal & Nichol, 2021), the standard equation for

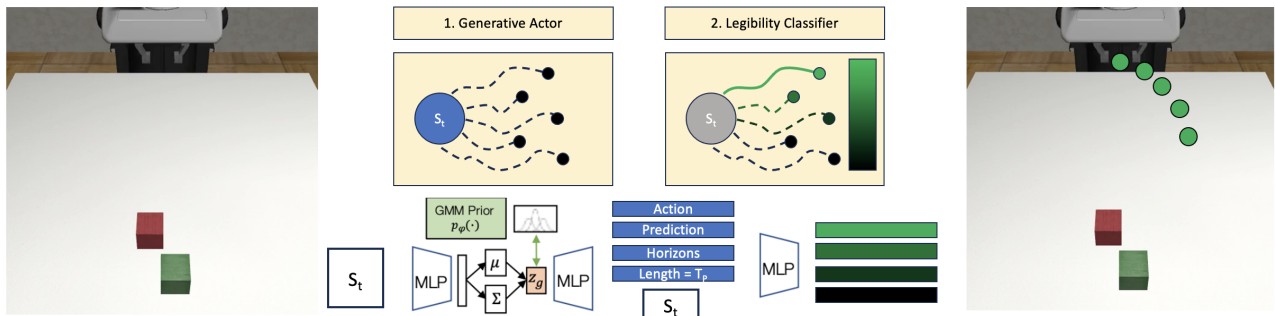

*Figure 1.* GLMM: In the first step, a VAE proposes diverse sub-trajectories from various modes in the data. A classifier than assigns a legibility score to each of the candidates based on how likely it is to see that sub-trajectory on a path to $g^*$. Finally we carry out the actions from the candidate with the highest score in order to get legible motion.

$\Delta_s \log P(s|g)$ is modified by scaling the conditioning term with $\gamma$.

$$\Delta_s \log p_\gamma(s|g) = \Delta_s \log P(s) + \gamma \Delta_s \log P(g|s) \quad (6)$$

By increasing the weight of the conditional term, we are able to amplify the influence of the class signal. Recent advances (Ho & Salimans, 2022) have made it more common to see classifier-free guidance as this eliminates the need to train a separate classifier on partially noised data. Classifier free-guidance combines a conditional and unconditional model with a $\gamma$ term that control their relative weights.

$$\Delta_s \log p_\gamma(s|g) = (1 - \gamma)\Delta_s \log P(s) + \gamma \Delta_s \log P(g|s) \quad (7)$$

When gamma is greater than 1, the class signal is amplified to a degree than is not possible with a standard conditional diffusion models.

Guidance has found a lot of success in the image domain because it generates higher quality samples at the cost of mode coverage. Visually, images produced with guidance will be sharper and have more class specific characteristics. Note that the class signal amplified by classifier guidance, $P(g|s)$, is essentially the same term maximized by legibility, $P(g|\xi)$. We propose that in the action generation domain, these class specific characteristics are the same characteristics that make motion legible.

In this paper we cannot directly use the equations for classifier guidance as we choose to use VAEs instead of diffusion models. We make this choice because of VAEs' relative ease of implementation and their inclusion in popular simulation libraries (Mandlekar et al., 2021). Additionally, our method of guidance is agnostic to the underlying generative model. We can implicitly implement guidance by jointly training a classifier and a generative model. The model produces a variety of outputs $X$ and the classifier selects the output $x \in X$ with the highest $P(g^*|x)$. This method will work as

long as the underlying generative model produces diverse, multi-modal outputs.

## 4. Method: GLMM

### 4.1. Overview

We first provide an overview of GLMM, our two stage method for generating legible motion (Figure 1), and describe how each step works at test time. The two phases involve a generative actor that proposes a diverse range of sub-trajectories and a goal-conditioned classifier that evaluates the legibility of each sub-trajectory. At state $s_t$, the actor will propose $N$ different action trajectories of length $T_p$: $\tau = (a_t, a_{t+1}, ..., a_{t+T_p})$. For each proposal $\tau$, the goal legibility classifier $f_l$ takes in $s_t$ and the goal $g^*$ as context and scores the proposal based on $P(g^*|s_t, a_t, ..., a_{t+T_p})$, which is the likelihood of $\tau$ being part of a trajectory that leads to $g^*$. For whichever proposal scores the highest, we then carry out $T_a$ actions from the length $T_p$ prediction horizon.

In our experiments we compare the performance of our algorithms to a goal conditioned VAE (G-VAE) and a VAE without goal conditioning. We use the Robomimic (Mandlekar et al., 2021) implementation of these two algorithms. Reference documentation for implementation details. Below we describe the generative actor and the goal-conditioned motion legibility classifier in detail.

### 4.2. Generative Actor

The generative actor that we use is a VAE trained on state plus actions pairs $(s_t, a_t, a_{t+1}, ..., a_{t+T_p})$ sampled from trajectories in the dataset. The VAE learns a conditional distribution $P(a_t, a_{t+1}, ..., a_{t+T_p}|s_t)$ to produce action horizons of length $T_p$. Note that the actor is conditioned on the current state, not the goal state. Because our data is

multi-modal, we adopt a learned Gaussian Mixture Model (GMM) prior $p_\phi(z) = \sum_{k=1}^{K} w_\phi^k \mathcal{N}\left(\mu_\phi^k, (\sigma_\phi^k)^2\right)$ in place of the standard Gaussian $\mathcal{N}(0, 1)$. To generate proposals, we independently sample $N$ times from the prior distribution. Once a proposed horizon is selected, we only roll out the first $T_a$ actions from the $T_p$ length sequence (this is done in open loop).

### 4.3. Goal-conditioned Legibility Classifier

The goal-conditioned legibility classifier is trained to predict $P(g^*|s_t, a_t, ..., a_{t+T_p})$. For this task we use a multi-layer perceptron (MLP) trained on trajectory sequences of length $T_p$ and a one-hot encoded goal vector $g_v$. These sequences are sampled from the training data and include the initial state $s_t$, the following $T_p$ actions $(a_t, ..., a_{t+T_p})$, and $g_v$ for each demonstration. The MLP has two hidden layers of dimensions 300 and 400, ReLU activation functions, and output dimensions equal to the number of goals. The final outputs are passed through a softmax function and the cross entropy loss is defined as $L_c = -\sum_{i=1}^{len(g_v)} g_v^i \log(p_i)$.

The classifier's learned probabilities serve as a legibility score for state and multi-action pairs. If a trajectory segment $\xi (s_t, a_t, a_{t+1}, ..., a_{t+T_p})$ is legible, then the classifier should be confident that it is headed towards $g^*$ (i.e. $p(g^*|\xi)$ is high). If we want legible motion towards goal $g^*$, we choose the proposed actions to which the classifier assigns the highest probability. This is similar to Equation 6 and Equation 7 when the influence of $P(g|s)$ is maximized. So while we cannot directly use those gradient based equations with a VAE, our technique implicitly provides classifier guidance.

## 5. Experiment

### 5.1. Task and Demonstration Data

We evaluate GLMM on a simulated pick-and-place task in which a Franka Panda Robot is required to pick up one of two offset blocks as in Figure 1. This is a common two-goal task for evaluating legibility and is similar to the setups in (Dragan et al., 2013; Zhao et al., 2020). We use Robosuite (Zhu et al., 2020) as our simulation environment and all demonstrations were captured by an expert demonstrator using a spacemouse from 3Dconnextion. A total of 280 demonstrations were collected that cover a variety of modes and a range of legibility. Visualizations of these trajectories can be seen in Figure 2. Our datasets are collected with low dimensional states, but future work will validate the results while using images as states.

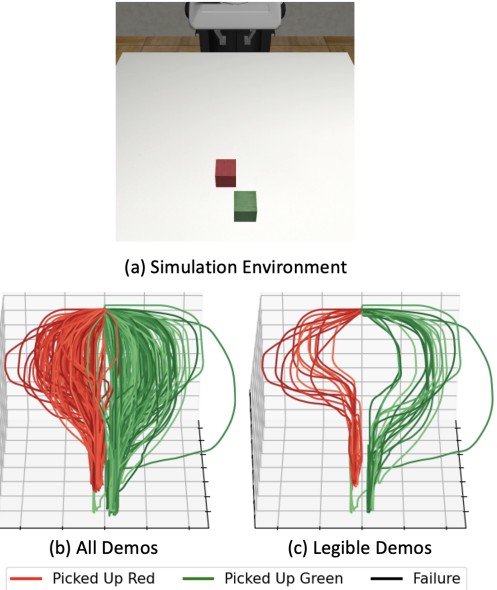

(a) Simulation Environment

(b) All Demos     (c) Legible Demos

— Picked Up Red    — Picked Up Green    — Failure

*Figure 2.* **Task Environment and Demonstrations:** (a) Pictured is the Robosuite (Zhu et al., 2020) simulation environment where we collected our demonstrations. There are two offset blocks and the task is completed once the robot arm picks one up. (b) This graph plots the robot's end effector position for all 280 demonstrations. (c) This graph plots the most legible trajectories in the dataset (top 10%) based on Equation 8.

### 5.2. Variables

In this experiment our independent variable is the algorithm used and our dependent variables are success rate and legibility. We tested three different algorithms on our pick-and-place task: GLMM, VAE (no goal conditioning), and G-VAE (goal conditioned VAE). Each algorithm was trained for 2,000 epochs, and the best performing checkpoint was used for evaluation. For these algorithms we set the prediction horizon $T_p = 10$ and the action horizon $T_a = 1$.

### 5.3. Evaluation Metrics

We evaluate the success rate and legibility of our algorithms after averaging across 100 rollouts of each algorithm.

To evaluate task success rate we measure the percentage of time the robot successfully picked up a block. This is counted as successful regardless of the color of the block. We formulate it this way because not all of our algorithms can be conditioned to pick up one block over the other. Additionally, in collaborative environments, all tasks need to be completed at some point. So reaching a goal, even if it is not the intended one, is not necessarily a failure.

For our task, we evaluate the legibility of the generated

*Table 1.* Success rate of generative algorithms taken after 100 task rollouts.

| ALGORITHM | SUCCESS RATE |
|-----------|--------------|
| VAE | **.97** |
| G-VAE | .93 |
| GLMM | .86 |

trajectories using a hand-designed metric. This metric measures the legibility for a complete trajectory $\xi_{s \to g^*}$ that goes from start state $s$ to target goal $g^*$. It takes into account the coordinates of all the non target goals $\{g \in G : g \neq g^*\}$ and the coordinates of every state $s_i$ included in $\xi_{s \to g^*}$ such that $s_{len(\xi)} = g^*$. We calculate euclidean distance in three dimensions. At each state in $\xi_{s \to g^*}$ the legibility contribution is weighted by $\frac{1}{i}$ such that more weight is given to earlier parts of the trajectory (as is common when computing legibility). This equation assigns a high legibility to states that are far from every non-target goal as it minimizes the probability that we are heading towards those goals (thus maximizing $P(g^*|\xi_{s \to s_i})$):

$$L(\xi_{S \to G^*}) = \sum_{s_i \in \xi_{S \to G^*}} \sum_{\{g \in G : g \neq g^*\}} \frac{||g - s_i||_2}{i} \quad (8)$$

This function combines Equations 2 and 3 by assigning higher legibility to trajectories that take longer to reach the non-target goal. We set the maximum legibility as $max_{\xi \in D} L(\xi)$ and minimum legibility as $min_{\xi \in D} L(\xi)$ where $D$ is our demonstration data. We normalize our reported legibility values to be within this range. Additionally, if the task is not successful, it is not evaluated for legibility and is dropped from the calculation of an algorithm's average legibility.

### 5.4. Hypotheses

- **Hypothesis 1:** If an agent is trained with GLMM, it will produce more legible trajectories than agents trained with VAE or G-VAE.

- **Hypothesis 2:** If an agent is trained using VAE, it will have a higher success rate than agents trained with G-VAE or GLMM.

- **Hypothesis 3:** If an agent is trained with GLMM, then it will produce the highest amount of failed rollouts

## 6. Results

The plots of the robot arm position for all 100 rollouts are shown in Figure 3. The success rate for VAE is the highest

*Table 2.* Average legibility of generative algorithm as measured by Equation 8. Legibility of 1 corresponds to maximum legibility in training data, legibility of 0 corresponds to minimum.

| ALGORITHM | AVERAGE LEGIBILITY |
|-----------|--------------------|
| VAE | $.24 \pm 0.06$ |
| G-VAE | $.25 \pm 0.11$ |
| GLMM | $\mathbf{.44 \pm 0.10}$ |

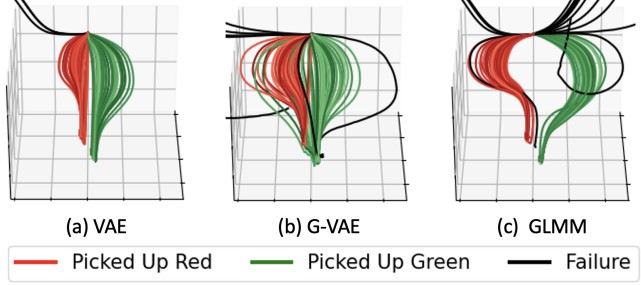

(a) VAE  (b) G-VAE  (c) GLMM

— Picked Up Red  — Picked Up Green  — Failure

*Figure 3.* **Trajectories From 100 Task Rollouts:** Each plot is of the robot arm's gripper position. (a) VAE generated trajectories (b) G-VAE generated trajectories (c) GLMM generated trajectories

at 97%, G-VAE at 93%, and GLMM performs the worst with 86% success rate. These results are shown in Table 1.

VAE has the lowest legibility at 24% and it also has the lowest variance. G-VAE and GLMM almost have the same variance, but GLMM has a much higher legibility at 44% compared to G-VAE at 25%. Note that the most legible trajectory in the training data, as evaluated by Equation 8, is defined as having 100% legibility. The least legible trajectory in the training data is defined to have 0% legibility. Legibility results are shown in Table 2 and Figure 4.

## 7. Discussion

**Hypothesis 1:** We found that the average legibility of GLMM is highest out of all three algorithms. Our implicit classifier guidance was able to select sub-trajectories $\xi$ that maximized $P(g^*|\xi)$. Because this term is known to maximize legibility, it's unsurprising that GLMM led to more legible trajectories. The fact that GLMM produces more legible trajectories than G-VAE suggests that our method for implicit classifier guidance is successful. We are able to increase the influence of class specific characteristics in the same manner as explicit classifier guidance.

**Hypothesis 2 and 3**: We find support for our second hypothesis as VAE has the highest succcess rate. We also find support for our third hypothesis as GLMM has the lowest success rate. When guiding the VAE with a classier, we force it to choose the **most** legible sub-trajectory that is

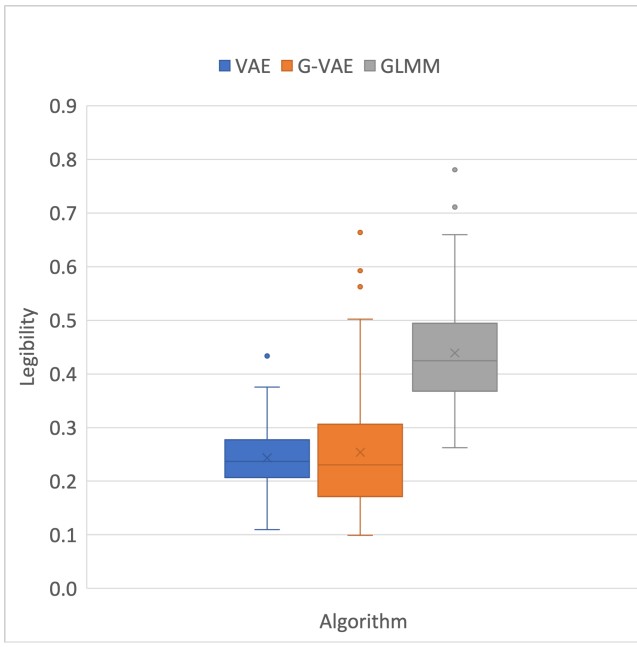

*Figure 4.* **Box and Whisker Plot of Algorithm Legibility:** Legibility is calculated using Equation 8 and reported as a percentage. The least legible trajectory in the training data is defined to have 0% legibility, the most legible is defined as having 100% legibility

proposed. This can be an issue because of overconfidence in out of distribution (OOD) states. Offline algorithms are known to struggle with OOD regions as they have not seen training data for these states. Without this training data, a classifier may score an OOD sub-trajectory as very legible because it's estimations are faulty in these regions. Once in OOD states, the agent will often suffer from compounding error issues and this will lead to task failure. Looking at the plotted trajectories from Figure 3, it is clear that GLMM suffers from these issues. In future studies, adding outlier protection such that we select the $n^{th}$ highest scoring sub-goal may help mitigate this issue.

Another related issue arises when following uncommon trajectories. Because the classifier is picking sub-trajectories that maximize $P(g^*|\xi)$, we will not follow paths that lead to multiple goals. If $\xi$ is seen in demonstrations going to the red block and demonstrations going to the green block, the classifier will give $\xi$ a low score. In our case, this means we avoid paths that go straight towards the goals. However, the path that leads to multiple goals can often be the most traveled. This is because many different demonstrations converge in this area. With lots of training data in these areas, we are better able to recover from errors. So if GLMM forces us down uncommon paths, we won't be able to recover from errors as effectively and will have a lower success rate. This is the reason why other papers

that explore classifier guidance in the context of imitating human behaviors have noted decreased model performance (Pearce et al., 2023). A main takeaway from this paper is that while classifier guidance may decrease success rate in the action generation domain, it can still be useful if you are concerned with generating legible motion.

VAE does not have to contend with any of these issues, hence our assumption that it would have the best task success rate. In the unconditioned form, the generative model is free to take the most common actions. G-VAE has some constraints on the actions it is trained to generate, so we believe this leads to the slightly decreased success rate compared to the unconditioned form.

## 8. Conclusion

We introduce GLMM, a framework for generating legible motion from multi-modal human demonstrations. **Our experiments show that while the implicit classifier guidance afforded by GLMM decreases task success rate, it does indeed optimize for legible trajectories**. Future work will confirm that as task complexity increases, GLMM continues to produce legible motion. We also plan to incorporate diffusion models into our experiments and exploring the effect of varying guidance weight. One limitation of our work is the assumption that the underlying generative model can capture multi-modal distributions. Further studies must be done to explore the extent to which generative models are able to accurately imitate the diversity of human demonstrations.

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

You can have as much text here as you want. The main body must be at most 8 pages long. For the final version, one more page can be added. If you want, you can use an appendix like this one, even using the one-column format.

