# OpenReview forum: "Legible Robot Motion from Conditional Generative Models"
_ICML.cc/2023/Workshop/ILHF — ILHF Workshop ICML 2023_

### Official Review · Reviewer_Tgts · 2023-06-15
**Review for Legible Robot motion from Conditional Generative Models**

**Rating:** 7
**Confidence:** 3

**Review:**

Summary: Extends legibility work for complex manipulation tasks by using generative models to capture highly unstructured data. Legibility is maximization of $P(g^{*}|\xi_{s\rightarrow q})$, where $g$ is the goal and $\xi_{s\rightarrow q}$ is a partial trajectory from $s\rightarrow q$. This method learns a classifier to rank sub-goals from the model on their legibility. The classifier learns $P(g|s)$ using a generative model trained to output the parameters of a conditional Gaussian from sampled trajectories. Goals are encoded as one-hot vectors to train the classifier with cross-entropy.

This work provides an interesting method for generating legible motion according to the first principles of mathematical legibility which is interesting and insightful. The use of a legibility classifier is interesting and appears to be novel.


This paper is somewhat limited in that it provides a classifier-guidance equation but then is limited in the choice of generative model (VAE instead of diffusion model). It certainly seems possible to use a diffusion model here.

The choice of a discrete set of goals seems to limit (necessary for the one hot representation), since the motivation comes from complex manipulation tasks, and goals representation in 3D space would leverage some of the structure of an environment.

For a work in legibility, the use of a hand-designed legibility measure is somewhat disappointing: it seems like the gold standard would be a human experiment demonstrating that the method provides a meaningful advantage to humans being able to predict the robot final goal. It would also be helpful to have a better illustration of why the legibility score is a good proxy for human legibility.

It seems plausible that there are some drawbacks to using a legibility classifier compared to a hardcoded method, such as sampling bias and data limitations especially as the goal space expands. Some analysis of this in experiments comparing classification with other legibility methods might be useful.

The work compares against VAE and G-VAE methods for generating trajectories, neither of which are actually legibility algorithms (these are just tools for generating a variety of trajectories). It is not clear from the problem setting that it would be impossible to use a hard-coded legibility method (such as used for pick and place) in the context of the actual experiments run.

---

### Official Review · Reviewer_f8hx · 2023-06-16
**Good paper on legible robot motion**

**Rating:** 7
**Confidence:** 3

**Review:**

This paper studies the legible robot motion problem. The authors proposed Generative Legible Motion Models (GLMM), which aims to learn legible trajectories from human demonstrations by conditional generative methods.

Quality: The paper is technically sound. While the experimental results are somewhat primitive, it still demonstrates the effectiveness of the proposed GLMM and provides valuable insight on the trade-off between legibility and task success rate.

Clarity: The paper is generally well-written and easy to follow.

Originality and Significance: The problem of generating legible motion is an important problem for our robotics community. A robots that acts in an understandable and interpretable way is critical for robot and human interaction. Given the conditional generative model 's capability of modeling distribution of unstructured data, the reviewer found the idea of using a conditional generative model to learn from human demonstration reasonable and convincing.

---

### Decision · Program_Chairs · 2023-06-20

Accept